

# Diagnosing weakly first-order phase transitions by coupling to order parameters

Jonathan D'Emidio[1]*, Alexander A. Eberharter[2] and Andreas M. Läuchli[3,4]

**1** Donostia International Physics Center, 20018 Donostia-San Sebastián, Spain
**2** Institut für Theoretische Physik, Universität Innsbruck, A-6020 Innsbruck, Austria
**3** Laboratory for Theoretical and Computational Physics,
Paul Scherrer Institute, CH-5232 Villigen, Switzerland
**4** Ecole Polytechnique Fédérale de Lausanne (EPFL),
Institute of Physics, CH-1015 Lausanne, Switzerland

* jonathan.demidio@dipc.org

## Abstract

The hunt for exotic quantum phase transitions described by emergent fractionalized degrees of freedom coupled to gauge fields requires a precise determination of the fixed point structure from the field theoretical side, and an extreme sensitivity to weak first-order transitions from the numerical side. Addressing the latter, we revive the classic definition of the order parameter in the limit of a vanishing external field at the transition. We demonstrate that this widely understood, yet so far unused approach provides a diagnostic test for first-order versus continuous behavior that is distinctly more sensitive than current methods. We first apply it to the family of $Q$-state Potts models, where the nature of the transition is continuous for $Q \leq 4$ and turns (weakly) first order for $Q > 4$, using an infinite system matrix product state implementation. We then employ this new approach to address the unsettled question of deconfined quantum criticality in the $S = 1/2$ Néel to valence bond solid transition in two dimensions, focusing on the square lattice $J$-$Q$ model. Our quantum Monte Carlo simulations reveal that both order parameters remain finite at the transition, directly confirming a first-order scenario with wide reaching implications in condensed matter and quantum field theory.



# 1 Introduction

The theory of phase transitions is an integral component in the understanding of many body phenomena, playing a significant role in fields ranging from statistical mechanics to condensed matter and high energy physics. The most recent advancements in this area have focused on phase transitions beyond the Landau-Ginzburg-Wilson (LGW) paradigm, where models have been proposed and studied extensively both analytically and numerically. In these scenarios the field theory describing the transition is not given in terms of the order parameters, as in the LGW paradigm, but instead is formulated in terms of fractionalized degrees of freedom. This leads to the possibility of a generic, *continuous* phase transition between two ordered phases whose symmetry groups and broken symmetries are not mutually compatible.

Prominent candidate examples for such exotic transitions are the 'deconfined quantum critical points' (DQCP) between Néel ordered antiferromagnetic and valence bond solid phases in quantum spin systems [1, 2]. The model believed to epitomize this scenario is the so called *J-Q* model [3] (to be discussed in detail below), which to date has been the most well studied in this context. Indeed, an impressive body of numerical work has been devoted to the analysis of the nature and the critical exponents of this and related models [3–17]. While most of the numerical data for the $S = 1/2$ *J-Q* model has been interpreted as evidence for a continuous quantum phase transition, albeit with significant corrections to scaling, some authors have however interpreted their data as evidence for a weakly first order transition. The current consensus opinion in the community is that the true nature of the *J-Q* transition remains to be settled definitively.

On the analytical side, several connections between the original (NCCP[1]) DQCP theory [1, 2] and other field theories of current interest such as the Abelian Higgs model (scalar $QED_3$) [18], the $SO(5)$ nonlinear sigma model with a Wess-Zumino term [19] or fermionic $QED_3$ [20] have been put forward [21]. It was first believed that the NCCP[1] theory is continuous, but Refs. [22–25] put forward and discussed scenarios of colliding fixed points in theory space, where the annihilation of two real fixed points provides a possible mechanism for pseudo-universal weakly first order behavior through the appearance of complex fixed points. In general, it is an important open question to determine the critical number $N_c$ of (bosonic or

fermionic) matter field flavors coupled to gauge fields, separating a regime of conformal field theories in the infrared (i.e. continuous transition behaviour) from a regime of pseudo-critical (weakly) first order regime in the infrared. This scenario also applies to the classical $Q$-state Potts model in two dimensions where the conformal window resides below $Q \leq 4$, resulting in (in)famously weak first order transitions near $Q > 4$ [26–28]. Additionally, similar ideas might also apply regarding the conformal window of non-abelian gauge theories [23], with possible implications for the hierarchy problem in particle physics.

It is therefore of great interest to refine numerical simulations so that conformal windows can be clearly resolved, which means developing highly sensitive methods for detecting weak first order transitions. However, pseudo-critical behavior implies a finite—but huge—correlation length at the transition. As a result, a conclusive diagnosis seemingly becomes impossible since the general wisdom is that system sizes with a linear extent at least as large as the correlation length are required to resolve the first order nature of the phase transition [29].

In this work we revive the textbook definition of the order parameter from the classical theory of phase transitions, which has all but fallen out of use in the numerical community. We find that applying these ideas in a modern context has the power to resolve weakly first order transitions with exquisite detail. After introducing the basic idea behind our approach in Sec. 2, we demonstrate the power of our method in the Hamiltonian formulation of the well-understood $Q$-state Potts model in Sec. 3 using an infinite matrix product state (iMPS) implementation, where we find distinct signatures for first order behavior manifesting at correlation lengths of *a few* lattice spacings, despite the correlation length at the transition itself reaching on the order of a thousand lattice spacings for $Q = 5$. We then move to two dimensional quantum critical phenomena using a Quantum Monte Carlo implementation, first in Sec. 4, where we show that this method corroborates the established continuous nature of the $O(3)$ Wilson-Fisher CFT quantum critical behavior of a family of explicitly dimerized $S = 1/2$ quantum magnets. Finally in Sec. 5, we address the Néel to valence bond solid phase transition in the square (and rectangular) lattice $S = 1/2$ $J$-$Q$ model and provide direct evidence for a first order scenario, thus resolving a long standing debate in the field, with implications in many directions both in condensed matter and in quantum field theory.

## 2 Outline of the method

In an ordinary symmetry breaking phase transition there are two complementary conceptual approaches to track the order parameter $\mathcal{O}$ as a function of a control parameter $g$ (temperature $T$ or a Hamiltonian parameter). We assume that $g > g_c$ is the symmetry broken phase, $g < g_c$ the paramagnetic phase with $g_c$ the transition point.

- In a symmetry preserving setup one measures the square of the order parameter $\langle |\mathcal{O}|^2 \rangle_g$ or functions thereof (such as Binder cumulants or order parameter susceptibilities) for finite systems and then extrapolates to infinite system size using finite-size scaling techniques.

- On the other hand, it has long been known that one can directly measure the order parameter by coupling it to a uniform external field via adding a term $\lambda \int \mathrm{d}^d x\, \mathcal{O}(x)$ to the Hamiltonian ($d$ denotes the space dimension). One then extrapolates $\langle \mathcal{O} \rangle_{g,\lambda}$ to infinite size at fixed coupling $\lambda$, then takes the limit as $\lambda \to 0^+$, yielding the order parameter $\langle \mathcal{O} \rangle_g$ in the thermodynamic limit.

In the symmetric setup if the transition occurring at $g_c$ is continuous then we expect the standard power law behaviour

$$\langle \mathcal{O} \rangle_g \sim (g - g_c)^\beta, \quad g \to g_c^+,$$

with $\beta$ an appropriate critical exponent, while in an external field at the critical point $g_c$:

$$\langle \mathcal{O} \rangle_{g_c, \lambda} \sim \lambda^{1/\delta}, \quad \lambda \to 0^+,$$

with $1/\delta$ a critical exponent which is controlled by the universality class of the transition at $g_c$.[1]

If however the transition at $g_c$ is first order then there is a coexistence of the paramagnetic and the symmetry broken phase at $g_c$. The applied field $\lambda$ then prefers the symmetry broken phase, leading to a finite $\langle \mathcal{O} \rangle_{g_c} \equiv m_c$, the (unique) value of the order parameter discontinuity at the transition.

The central quantity of interest in our work is the following logarithmic derivative at the critical coupling $g_c$:

$$[1/\delta](\lambda) := \frac{\partial \log \langle \mathcal{O} \rangle_{g_c, \lambda}}{\partial \log \lambda}. \tag{1}$$

We also refer to this quantity as the "running exponent $1/\delta$", since we will study this quantity as a function of $\lambda$. According to the discussion above we expect $[1/\delta](\lambda)$ to approach $1/\delta$ for $\lambda \to 0^+$ in the continuous case, while the finite value value $m_c$ of the order parameter at $g_c$ in the first order case drives the logarithmic derivative to zero.

While not of central interest for the present work, one can also track the behavior of $[1/\delta](\lambda)$ for other values of $g$. In the symmetry broken regime $g > g_c$ we expect the running exponent to scale to zero, as in the first order case at $g_c$. The paramagnetic phase $g < g_c$ requires some more care. For a unique, gapped ground state in the paramagnetic phase we expect a standard linear response regime, resulting in a running exponent $[1/\delta](\lambda) = 1$ for small fields $\lambda$. This is actually also the expected behaviour for all $g$ in a generic finite size system at very small $\lambda$. We will witness this phenomenon for the finite size quantum Monte Carlo simulations presented in Secs. 4 and 5.

While in the early days of Monte Carlo investigations of phase transitions approaches with and without a coupling to the order parameter were pursued [30], the tediousness of the double limit with an external field put this method at a disadvantage compared to symmetric setups that could locate critical points and measure exponents with fewer simulations. Subsequently with the development of powerful cluster algorithms that are tailor made for symmetric models [31,32], the order parameter coupling approach has seemingly fallen into complete disuse in modern numerical simulations of phase transitions. A notable exception is the pinning field method used in Ref. [33] to resolve the nature of the semimetal to Mott insulator transition in the honeycomb Hubbard model. In that work the applied field was confined to a single site, whereas we use a spatially extended coupling to the order parameter in our work.

In our present work we demonstrate that the order parameter coupling approach is a powerful tool for diagnosing weak first order transitions, performing well beyond the abilities of the currently used symmetric approaches. For our purposes we find its apparent drawbacks to be inconsequential in practical simulations, allowing it to be seamlessly integrated into state of the art numerical algorithms, here using infinite matrix product state (iMPS) and finite size quantum Monte Carlo (QMC) algorithms. In fact the presence of an order parameter coupling allows us to devise statistically exact QMC estimators for the running exponents as a function of the external field, eliminating finite-difference errors and facilitating the diagnosis of first order transitions. In the next section we demonstrate for the Q-state Potts model family that the difference in behavior of Eq. (1) between continuous and weakly first order transitions is surprisingly stark, and occurs at rather large values of the order parameter coupling $\lambda$ and correspondingly short correlation lengths.

---

[1]The exponent $1/\delta$ can be obtained for a conformal field theory from the order parameter scaling dimension $\Delta_{\mathcal{O}}$ and the space-time dimension $D$ as $1/\delta = \Delta_{\mathcal{O}}/(D - \Delta_{\mathcal{O}})$.

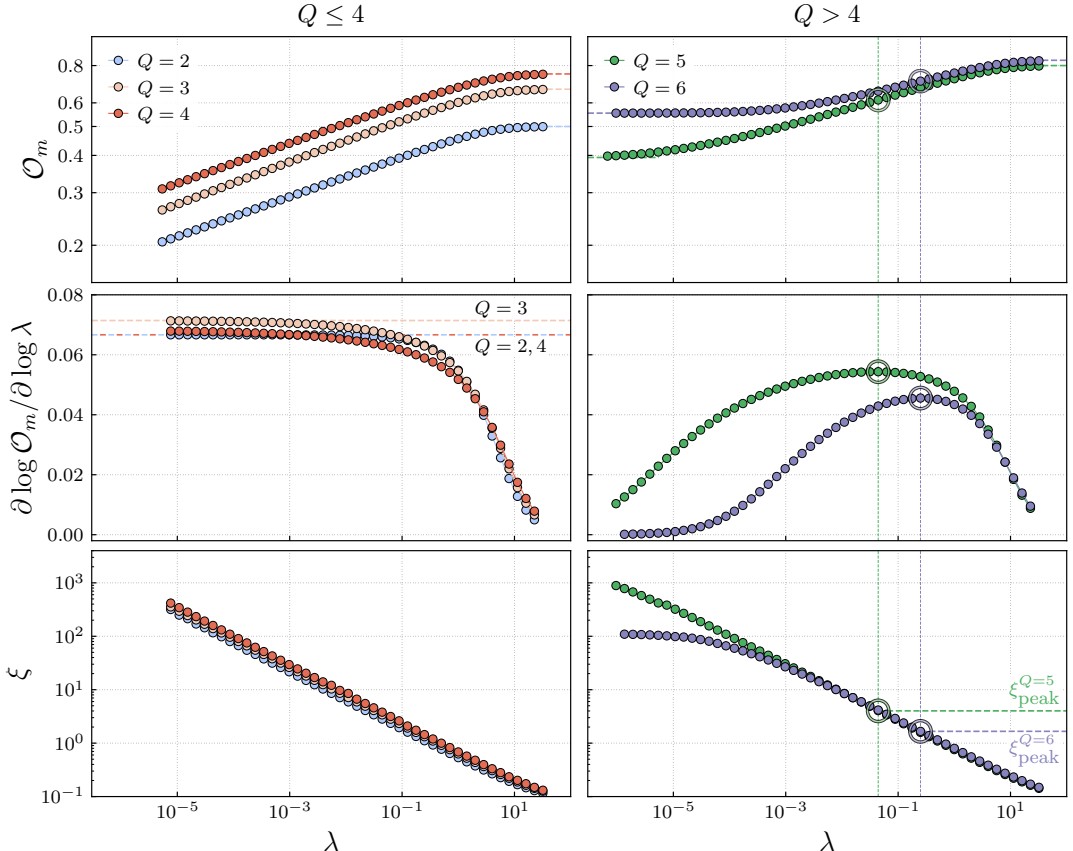

Figure 1: Quantum $Q$-state Potts Chain: infinite system MPS simulations for the quantum critical point with an applied symmetry breaking field $\lambda$. The left (right) column displays the exactly known continuous (weakly first order) transitions for $Q = 2, 3, 4$ ($Q = 5, 6$). The top row shows the order parameter $\mathcal{O}_m$ as a function of the perturbing field $\lambda$ on log-log axes. The inflection points in the right panel are highlighted with circles. The middle row presents the logarithmic derivative $[1/\delta](\lambda)$, i.e. the running exponent defined in Eq. (1), of the first row, highlighting the convergence towards the expected exponents in the left panel, and the existence of distinct maximum in the weakly first order cases in the right panel. In the bottom row we plot the extracted MPS correlation length, showing that the inflection point and the corresponding maximum in the running exponent for the weakly first order instances occur at correlation lengths of only a few lattice spacings.

## 3 The Q-state Potts model

We start discussing our method by an application on a challenging problem, the $Q$-state Potts model in the 1+1D Hamiltonian formulation [28, 34, 35], which basically corresponds to a spatially anisotropic version of the widely known 2D classical Potts model. We consider the Hamiltonian already fine tuned to the exact value of the quantum phase transition between the ordered and paramagnetic phase (i.e. working at $g_c$), and add a symmetry breaking field $\lambda$ favoring one of the $Q$ ferromagnetically ordered states, here chosen as $q = 0$:

$$H_{\text{Potts}}^\lambda = -\sum_i \sum_{q=0}^{Q-1} Q \, |q_i \, q_{i+1}\rangle\langle q_i \, q_{i+1}| - \sum_i M_i^x - \lambda \sum_i |0\rangle\langle 0|_i \,, \tag{2}$$

where $i$ runs over the sites of the chain and $q$ over the $Q$ distinct local states $q \in \{0, ..., Q-1\}$.

The operator $M_i^x$ has unity matrix elements between any two local spin states on site $i$.

In zero external ("longitudinal") field ($\lambda = 0$) this model sits exactly at a continuous quantum phase transition for $Q = 2, 3, 4$ that becomes discontinuous for (integer) $Q > 4$ [26]. Just above this threshold $Q > 4$ the first-order discontinuity is exceptionally weak and it is, in fact, notoriously difficult for Monte Carlo simulations to correctly identify the nature of the transition, because of so called pseudo-critical behaviour associated to large remnant correlation lengths at the first order transitions [27, 28]. This pseudo-critical behaviour is attributed to a fixed point collision in theory space, with complex fixed points arising for $Q > 4$. We now show that by introducing the external field $\lambda$ at the transition, a striking qualitative difference can be observed between the continuous and first-order cases at rather large couplings $\lambda$ and correspondingly short correlation lengths.

In the top row of Fig. 1 we display the zeroth-component magnetization $\mathcal{O}_m = \langle |0\rangle\langle 0|_i - 1/Q \rangle_\lambda$ as a function of the external field $\lambda$ that are obtained directly in the thermodynamic limit using an infinite system Matrix Product State (iMPS) DMRG algorithm, see App. A for details on the technical aspects. In the left column we display results for the known continuous cases $Q = 2, 3, 4$, while in the right column results for the weakly first-order cases $Q = 5, 6$ are shown. The $Q = 2, 3, 4$ data in the top row exhibits rather clean power law scaling of $\mathcal{O}_m$ as the coupling $\lambda$ goes to zero, while in the $Q = 5, 6$ cases a saturation of $\mathcal{O}_m$ to a non-zero residual magnetization is observed in the same limit. These limiting values are in very good agreement with the exact results obtained by Baxter [36]. In the middle row we calculate numerical finite-difference derivatives of $\log \mathcal{O}_m$ with respect to $\log \lambda$ in order to highlight possible power law behavior $\mathcal{O}_m \propto \lambda^{1/\delta}$. Indeed at small $\lambda$ the derivatives for $Q = 2, 3, 4$ approach the expected values for the known CFTs governing the fixed points: $1/\delta = 1/15$ for $Q = 2, 4$ and $1/\delta = 1/14$ for $Q = 3$. Note for $Q = 4$ there is a known logarithmic correction to the power law behavior: $m \propto (\lambda/\log \lambda)^{1/15}$ [27], leading to a tiny non-monotonicity for $Q = 4$. In the weakly first order cases $Q = 5, 6$ in contrast we observe a pronounced maximum in the derivatives, which has its origin in the inflection point in the original data, c.f. top panel. For our model and $Q > 4$ there is a saturation in $\mathcal{O}_m$ both for small and large[2] values of $\lambda$ leading necessarily to (at least) one inflection point at an intermediate value of the external field, which we denote by $\lambda^\star(Q)$. In order to assess at what length scales the pronounced feature of a maximum and the downward drift to zero at smaller $\lambda$ occurs, we present in the bottom row of Fig. 1 the correlation lengths $\xi^Q(\lambda)$ obtained from the transfer operator of the iMPS wave function for the different values of $Q$. In the continuous cases $Q = 2, 3, 4$ we observe again power-law behavior as expected. The most notable observation for the weakly first order cases $Q = 5, 6$ is that the correlation length $\xi^Q_{\text{peak}}$ measured at the coupling $\lambda^\star(Q)$ is actually quite small, i.e. about 4 resp. 2 lattice spacings for $Q = 5$ and $Q = 6$ respectively. These correlation lengths $\xi^Q_{\text{peak}}$ are two to three orders of magnitude smaller compared to the huge, albeit finite, correlation lengths at the first order phase transition itself [37, 38].

We believe that the pronounced maximum feature of $[1/\delta](\lambda)$ at intermediate values of the coupling $\lambda$ and the subsequent drift towards zero as $\lambda \to 0^+$ is a robust phenomenon for weakly first order transitions more generally, and it might have its origin in the colliding fixed point scenario advocated for the $Q > 4$ Potts models. It is notable that the very weak first order transition for $Q = 5$ has a broader maximum and a relatively slow approach to zero compared to the case $Q = 6$. It is however striking how different $Q = 5$ behaves in contrast to the continuous transition at $Q = 4$, despite the presence of a logarithmic correction in the latter case, usually spoiling a clean analysis.

These remarkable observations now pave the way to study many open problems in various fields where weakly-first order transitions are hard to discriminate from continuous phase transition with the methods available so far. As an important open question we will address

---

[2]at large values of $\lambda$ the observables saturate at $1 - 1/Q$ for all $Q$.

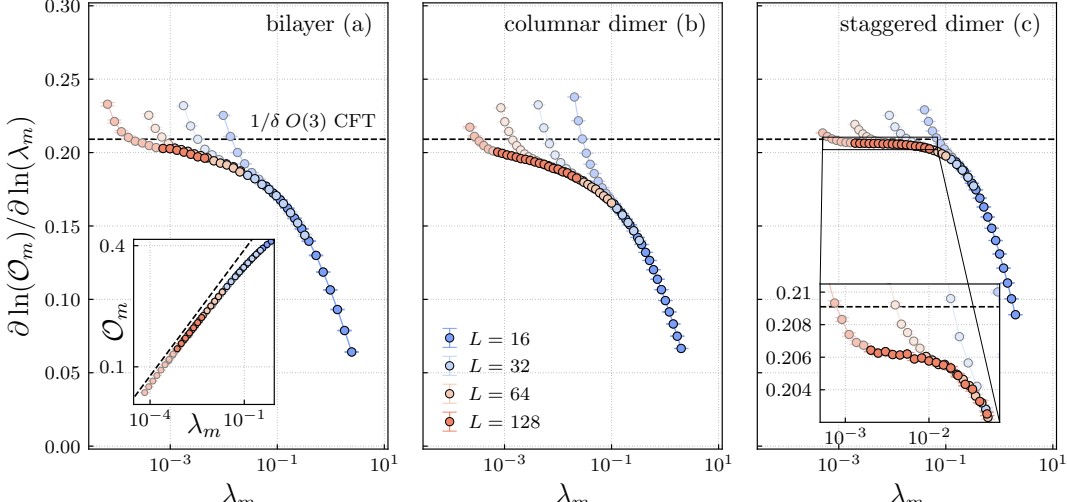

Figure 2: The order parameter (staggered magnetization) exponent $[1/\delta](\lambda_m)$, i.e. the running exponent defined in Eq. (1), for three lattice models realizing the $O(3)$ quantum phase transition: the Heisenberg bilayer, the columnar dimer model, and the staggered dimer model. In the left inset we provide the order parameter as a function of the external Néel field on log-log axes, showing clean power law scaling at low fields, where the expected power law $1/\delta = 0.2091(1)$ [39] is shown for reference with the dashed line. The measured running exponents in all three cases monotonically approach the expected value, behaving analogously with the continuous $Q = 2, 3, 4$ Potts model. We note that the staggered dimer model seems to show an initial fast approach to the O(3) exponent, followed by a slower approach at low fields (right inset). Here we have faded points that we roughly judge by eye to be in the finite size regime.

the nature of the phase transition in the *J-Q* model, which is a candidate for a DQCP. Before tackling this problem, however, let us first validate our approach for a family of 2+1D quantum many body systems with an undisputed continuous quantum phase transition, which we now study using a finite size quantum Monte Carlo method.

## 4 Quantum models for the $O(3)$ transition

We consider three models that host a quantum critical point in the 3D $O(3)$ universality class. The first is the well studied square lattice $S = 1/2$ Heisenberg bilayer system, whose Hamiltonian is given by

$$H_{\text{bi}}^m = J_1 \sum_{\langle ij \rangle} \sum_{a=1,2} \vec{S}_{ia} \cdot \vec{S}_{ja} + J_2 \sum_i \vec{S}_{i1} \cdot \vec{S}_{i2} + \lambda_m \sum_{i,a} (-1)^{r_i^x + r_i^y + a} S_{ia}^z . \tag{3}$$

Here $J_1$ couples nearest neighbor spins within each square lattice, and $J_2$ is the coupling between the layers. We have also added an external field $\lambda_m$ that couples to a component of the order parameter, in this case the staggered $S^z$ magnetization.

With $J_1, J_2 > 0$ this model undergoes a transition from a Néel ordered antiferromagnet for $J_2/J_1 < g_c$ to a dimer singlet phase on the interlayer bonds when $J_2/J_1 > g_c$, with $g_c = 2.52181(3)$ [40]. In order to probe the critical scaling at the transition, we compute the order parameter $\mathcal{O}_m = \langle S_{11}^z \rangle_{L,\lambda_m}$ on finite size systems of side length $L$ using the stochastic

series expansion (SSE) algorithm [41] with directed loops [42]. We furthermore have developed a statistically exact Monte Carlo estimator to directly measure the logarithmic derivative $\partial \log \mathcal{O}_m / \partial \log \lambda_m$ that eliminates finite difference errors (see Appendix B), cleanly extracting the running exponent $(1/\delta)$ as a function of the field $\lambda_m$.

In order to paint a more comprehensive picture we also study two other models that host the same $O(3)$ transition but with significant finite-size corrections to critical scaling [43] in one of them. Both models are taken on the square lattice (single layer), where again two different bond strengths $(J_2 > J_1)$ are used. Following the nomenclature of [43], we study the columnar dimer model (CDM), consisting of columns of x-oriented strong bonds with a Fourier component $(\pi,0)$, as well as the staggered dimer model (SDM), consisting of alternating x-oriented strong bonds with a Fourier component $(\pi,\pi)$. For the CDM and SDM we use the critical coupling ratios $J_2/J_1 = 1.90951$ and $J_2/J_1 = 2.51943$, respectively [43].

In Fig. 2 we show all three of the running Néel exponents $[1/\delta](\lambda_m)$ (and the bilayer Néel order parameter in the left inset) at the critical point for the three different models. In the bilayer model we have used $J_2/J_1 = 2.5223$, where the difference with the $g_c$ quoted above is inconsequential for this plot but shows better agreement with the O(3) exponent in finite size data collapses (see Appendix G). We have used $\beta = 2L$ for the bilayer model and $\beta = L/2$ for the CDM and SDM as in [43], all in units with $J_1 = 1$ (see Appendix F showing negligible temperature effects).

The bilayer order parameter (left inset) as well as the CDM and SDM order parameters (not shown) all display clean power law scaling at low fields, where the dashed line shows the expected power law for reference $1/\delta = 0.2091(1)$ [39]. The measured running exponents (main panels) provide a more fine-grained view of the approach to the expected power law as the field is lowered, where the $O(3)$ exponent is again plotted as a dashed line. In the finite size setup we are using, there is an $L$-dependent crossover scale for $\lambda_m$, below which one ultimately observes $\mathcal{O}_m \propto \lambda_m$, a generic result for any finite-size system in the limit $\lambda_m \to 0$. This phenomenon explains why the derivative curves for a given $L$ start to bend upwards at small $\lambda_m$. As the system size increases, this finite size regime is pushed to smaller values of $\lambda_m$ and a consistent picture representative for finite $\lambda_m$ at $L \to \infty$ emerges. The infinite size (and zero temperature) converged data reveals a monotonic increase of the running exponents, which approaches the $1/\delta$ value expected for a 3D $O(3)$ Wilson-Fisher universality class [39].

Remarkably, even in the SDM, where sizeable corrections to scaling and non-monotonicity of finite-size effective exponents have previously been reported [43, 44], we observe a clean monotonic approach to the expected exponent. We note that within our numerical resolution the SDM running exponent seems to contain a regime of fast approach at higher fields, giving way to a much slower approach at lower fields. Although a more careful scaling analysis of the SDM would be desired in this context, we can clearly see the broad picture that is captured by all three of these critical models. While the comparatively slow convergence of $[1/\delta](\lambda_m)$ towards the expected $1/\delta$ value renders our approach less useful to accurately determine $1/\delta$, we emphasize that the important result here is the *absence* of a pronounced maximum in $[1/\delta](\lambda_m)$ and the subsequent lack of a drift towards zero as $\lambda_m \to 0^+$.

This demonstrates that implementing our method using a finite-size QMC method for a well understood continuous 2+1D quantum phase transition leads to behavior in clear analogy to the *continuous* phase transitions in the $Q = 2, 3, 4$ Potts cases studied in 1+1D with iMPS.

# 5 The $J - Q$ models

Finally we turn to a main objective of this work, which is to shed light on the nature of the quantum phase transition between Néel order and valence bond solid (VBS) order in two di-

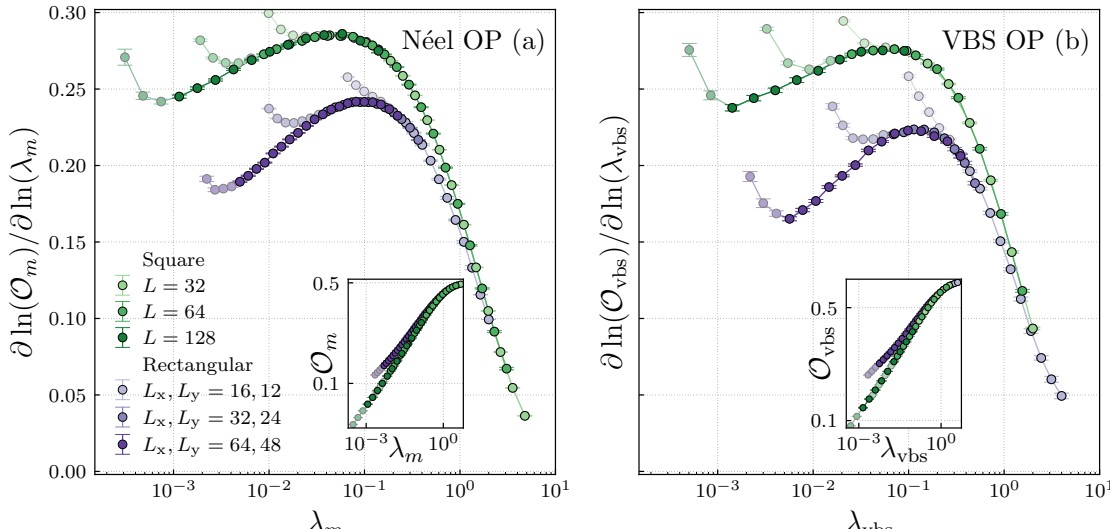

Figure 3: The Néel and VBS order parameter running exponents for the square and rectangular lattice $J$-$Q$ models tuned at the transitions $J/Q = 0.0447$ and $J_x/Q_x = 0.205$, respectively. For the square lattice we use $\beta = L/2$ ($Q = 1$) and for the rectangular lattice $\beta = L_x/2$ ($Q_x = 1$). The running exponents show a local maximum and persistent drift at low fields, behaving as the $Q = 5, 6$ Potts model. We observe a striking similarity between the known first-order rectangular case and the square lattice case, providing compelling evidence that the transition remains weakly first-order in the square lattice case as well.

mensional $S = 1/2$ spin systems, thought to be described by the deconfined criticality scenario. An important difference to the previously discussed cases is that the deconfined criticality scenario describes the transition between two ordered phases, therefore we have to track the behaviour of two separate order parameters at the transition point. At a continuous phase transition we expect both running exponents to approach their corresponding values dictated by the universality class in question. In contrast, at a first order phase transition the two order parameters are both expected to be finite, as the two ordered phases coexist at the transition point.

The most well studied model in this context is referred to as the $J$-$Q$ model [3], written as

$$H_{JQ} = J \sum_{\langle ij \rangle} (\vec{S}_i \cdot \vec{S}_j - \tfrac{1}{4}) - Q \sum_{\langle ijkl \rangle} (\vec{S}_i \cdot \vec{S}_j - \tfrac{1}{4})(\vec{S}_k \cdot \vec{S}_l - \tfrac{1}{4}). \tag{4}$$

Here $J > 0$ is the antiferromagnetic coupling between nearest neighbors $S = 1/2$ spins on a square lattice, and $Q > 0$ is a product of two adjacent $J$ terms acting on an elementary square of four spins in both the $x$ and $y$ orientations. When $Q = 0$ we are left with the Heisenberg model which has Néel order, and conversely when $J = 0$ the spins form a columnar VBS phase [3]. At a small value of the coupling ratio $J/Q \approx 0.04$, the transition between these seemingly unrelated orders takes place. Currently, the true nature of the transition in the $J$-$Q$ model is still under debate. While Refs. [3–5, 7–9] and [6, 16] interpret their data as being in favour of a continuous quantum phase transition, it appeared that the extracted critical exponents show pronounced drifts as a function of the maximal system size. However in these simulations no direct evidence for first order behaviour has ever been seen, such as a negative Binder cumulant or multiple peaks in histograms of the energy or order parameters. There are however some papers claiming to observe first order behaviour using the flowgram method [10–12].

To probe the nature of this transition we perform two separate studies: one in which a staggered magnetic field coupling to the Néel order parameter is added, and another with a

field coupling to the VBS order parameter. As with the O(3) models before, the $J$-$Q$ model with the Néel field is written

$$H_{JQ} + \lambda_m \sum_i (-1)^{r_i^x + r_i^y + 1} S_i^z \,,$$

with the observable $\mathcal{O}_m = \langle S_1^z \rangle_{L,\lambda_m}$. We write the model with the VBS field as

$$H_{JQ} + \lambda_{\text{VBS}} \sum_{\langle i,j \rangle \in \hat{x}-\text{even}} (\vec{S}_i \cdot \vec{S}_j - \tfrac{1}{4}) \,,$$

which preferentially selects one of the four columnar VBS patterns. The VBS order parameter is computed as the expectation value of the difference between even and odd x-bonds

$$\mathcal{O}_{\text{VBS}} = \langle \vec{S}_2 \cdot \vec{S}_3 - \vec{S}_1 \cdot \vec{S}_2 \rangle_{L,\lambda_{\text{VBS}}} \,.$$

Again, we have developed statistically exact QMC estimators for the logarithmic derivatives in both models (see Appendix B).

We furthermore compare the behavior of the $J$-$Q$ model to a known first-order Néel-VBS transition that is realized by introducing rectangular lattice anisotropy, as was previously studied in [16]. Following this methodology, we take spatially anisotropic couplings $J_y/J_x = 0.8$ and $Q_y/Q_x = 0.8$ on rectangular lattices with $L_x = 4L_y/3$. In this model we do not have a prior estimate of the transition, so it was located by scanning the binder ratio of the staggered magnetization (in zero field) for several system sizes (see Appendix E). Here we find a rough estimate of the transition, in this case $J_x/Q_x \approx 0.205$, is more than enough precision for the results we present here. Just as in the square lattice $J$-$Q$ model, we sit at the transition and introduce separate Néel and VBS fields while measuring the running exponents.

In Fig. 3, we show the running Néel and VBS exponents as in Eq. (1) for both the square lattice and rectangular lattice $J$-$Q$ models tuned to their respective transitions. For the square lattice $J$-$Q$ model we have used the transition value $J/Q = 0.0447$ [7]. The left panel presents data for each model coupled to the Néel order parameter field, while the right panel similarly presents data for both models coupled to the VBS order parameter field. We clearly observe strong deviations from critical power law scaling for both models with both effective exponents drifting toward zero at small field values, suggesting coexisting Néel and VBS order at the transitions. While our available system sizes do not allow us to track the running exponents all the way to zero, they nevertheless approach closely (surpassing, in the rectangular case) the unitarity bound for scalar operators in a 2+1D CFT $\Delta_\phi \geq 1/2$ [45], yielding a lower bound $1/\delta \geq 1/5$. The downward drift of the running exponents is substantial and the contrast to the behavior observed in the continuous O(3) models shown in Fig. 2 on the same vertical scale is striking. We further emphasize the similarity of the running exponents between the known first-order rectangular case and the square lattice case, as well as point out the resemblance to the behavior observed in the $Q = 5, 6$ Potts model, painting a compelling picture that the square lattice $J$-$Q$ model is weakly first order.

## 6    Discussion and outlook

Working tangentially to the current symmetry-preserving studies of quantum phase transitions by reintroducing the classic definition of the order parameter in a modern context, we have pushed the sensitivity to diagnose weakly first-order transitions to an unprecedented level. As an important application we have shown that the $SU(2)$ $J$-$Q$ model on the square lattice does not host a genuine DQCP, but instead a weakly first order transition with coexisting Néel and

VBS order at the quantum phase transition. This important result also corroborates recent field theoretical arguments claiming an absence of a genuine DQCP with $SO(5)$ symmetry in 2+1D [24,25,46] and validates early numerical simulations claiming first order behavior based on a flowgram analysis [10–12]. Furthermore it puts the $J$-$Q$ model on the same footing as a 3D classical loop model studied in [22], which shows also indications for a weak first order transition, and is expected to realize yet another lattice version of the same NCCP$^1$ field theory as the $J$-$Q$ model studied here [1,2].

In view of these results, it is clear that many previously studied models using similar methods need to be revisited [13,16]. As an important next step, one can determine where the critical window as a function of $N$ begins in the $SU(N)$ $J$-$Q$ models on the square lattice [4,6,15,16]. Our results might also have implications for phase transitions out of Dirac spin liquids by virtue of a conjectured duality [21].

Since our approach effectively allows for a controlled study as a function of the correlation length at the transition, it should naturally find applications in 2D tensor networks with applied perturbations (see [47,48] for studies of the 2+1D Ising model and a coupled Heisenberg spin ladder at criticality) to probe the existence of DQCP in frustrated quantum magnets [49,50], where QMC is not applicable due to the sign problem. A closely related important study now within reach is to probe the existence of $SU(N)$ Dirac Spin Liquids which are conjectured to exist in many frustrated spin models [51–55], and whose field theoretical description is fermionic QED$_3$ with $N_f = 2N$ massless fermion flavors. In the $SU(2)$ Dirac spin liquid context natural perturbations are related to fermion bilinears and monopole operators recently characterized for various lattices in Ref. [56].

The fact that in our approach moderate lattice sizes are typically sufficient to detect weak first order behaviour suggests an immediate applicability for fermionic determinantal QMC methods which typically operate at smaller system sizes compared to the QMC methods used in this work. Exotic quantum phase transitions related to those discussed in the present work have been reported for interacting fermion systems and might warrant an independent confirmation using our technology.

On a more speculative note it will be worthwhile to explore the possibility to transport the ideas developed and demonstrated in this work to lattice field theory simulations of QED, QCD or related theories of importance to high energy physics.

The DMRG and QMC source code as well as the data and plotting scripts for the main figures (Figure 1 to 3) are freely available online [57].

# Acknowledgements

We thank F. Alet, T.C. Lang, A. Nahum, S. Rychkov for valuable feedback on an earlier version of the manuscript. JD thanks M.S. Block for correspondence about the rectangular $J$-$Q$ model.

**Funding information** AAE and AML acknowledge support by the Austrian Science Fund (FWF) through Grant No. I 4548. JD acknowledges funding from the Spanish MCI/AEI/FEDER through grant PGC2018-101988-B-C21. The computational results presented have been achieved in part using the Vienna Scientific Cluster (VSC) and the LEO HPC infrastructure of the University of Innsbruck. Additional QMC simulations were performed using the facilities of the Scientific IT and Application Support Center of EPFL on the FIDIS cluster.

# A   iMPS for Potts model

In this appendix we present complementary information regarding the iMPS study of the one-dimensional $Q$-state quantum Potts model presented in Sec. 3.

We define the $Q$-state Potts model on a spin chain of $N$ sites and $Q$ spin states per site denoted as $|q_i\rangle$ with $q_i \in \{0, 1, \ldots, Q-1\}$. Writing the Hamiltonian in the eigenbasis of the interaction term we get

$$H^{\text{Potts}} = -\sum_{i=1}^{N}\sum_{k=1}^{Q-1}\left(K_i^k K_{i+1}^{Q-k} + (1+g)T_i^k\right) + \lambda\,|0\rangle\langle 0|_i\,, \tag{A.1}$$

where $K_i$ is a diagonal matrix with its eigenvalues being the $Q$-th roots of unity, $K_i\,|q_i\rangle = e^{i2\pi q_i/Q}\,|q_i\rangle$, and $T$ being the spin-flip operator: $T_i^k\,|q_i\rangle = |(q_i + k)\bmod Q\rangle$. The perturbation strength $\lambda$ couples to the sum over the projectors onto a single local state, here the $|0\rangle$-state. It is customary to also add a perturbation with a transversal field with coupling strength $g$ to the Potts Hamiltonian, for $g = 0$ the Hamiltonian (A.1) minus an energy density of 3 is equal to the Hamiltonian given in (2). At $\lambda = 0$ the Hamiltonian is invariant under the global action of the symmetric group $S_Q$ and it exhibits two phases. For $g < 0$ the system is ordered and the ground state space is $Q$-fold degenerate with the ground state breaking the $S_Q$ symmetry. When $g > 0$ the system is disordered and the non-degenerate groundstate preserves the symmetry. These phases are separated by a phase transition at $\lambda = 0$ and $g = 0$ which is of second order for $Q \leq 4$. For $Q = 2$ the Potts Hamiltonian reduces to the transverse field Ising model. In the following section we assume that $g = 0$.

To calculate the properties of its ground state for arbitrary $Q$ and $\lambda$ in the thermodynamic limit we employ the generalization of the Density Matrix Renormalization Group algorithm to infinite spin chains (iDMRG) [58]. It works by decomposing the state into a finite number of rank 3 tensors which are repeated infinitely along the chain and thus form the unit cell. In the present paper only unit cells of size two are used. This set of tensors is called an infinite Matrix Product State (iMPS) which allows us to approximate the state of the system by introducing a cutoff of the tensors' bond dimension $\chi$. The approximation limits the amount of entanglement contained in the state with the entanglement entropy being capped at $S = \ln(\chi)$. For systems at a quantum critical point the entanglement entropy of the ground state diverges, however since we are perturbing the critical systems with a relevant field the Hamiltonian is gapped, and for large enough bond dimension our iMPS representation is basically numerically exact. The computational difficulty increases with increasing correlation length, i.e. with smaller values of $\lambda$.

A necessary requirement for the iDMRG is an efficient Matrix Product Operator (MPO) representation of the Hamiltonian which is easy to achieve for models with only nearest-neighbour interactions. The energy expectation value of an MPS is then expressed as a tensor contraction of MPS and MPO. The algorithm variationally minimizes the energy by sweeping through the system, optimizing the tensors of 2 neighboring sites at a time, until the energy and entanglement entropy converge. In every sweep the eigenvalue problem is projected into these 2 sites and solved using the Lanczos algorithm which is based on calculating the action of the projected Hamiltonian on a wavefunction many times.

In order to speed up the tensor contractions we make use of the Hamiltonian's symmetry. At $\lambda = 0$ the $Q$-state Potts model is invariant under the global action of the non-abelian symmetric group $S_Q$, for $\lambda \neq 0$ this symmetry is reduced to its subgroup $S_{Q-1}$. It is technically much easier to deal with abelian groups thus we only consider the $Z_Q$ and $Z_{Q-1}$ subgroups respectively. In order to exploit the Hamiltonian's symmetry it needs to be decomposed in the right way. To

achieve this we apply a global on-site unitary transformation: $|\tilde{n}_i\rangle = U^{(Q)}|n_i\rangle$, with

$$U^{(Q)} = \begin{pmatrix} 1 & 0 \\ 0 & \tilde{U}^{(Q-1)} \end{pmatrix}, \tag{A.2}$$

where the $(Q-1) \times (Q-1)$ matrix $\tilde{U}^{(Q-1)}$ is defined by $\langle m'|\tilde{U}^{(Q-1)}|m\rangle = (Q-1)^{-1/2}\, e^{i\frac{2\pi}{Q-1}mm'}$. Note that the zeroth-component is unchanged: $|\tilde{0}\rangle = |0\rangle$. In the $Z_Q$ case, which is not relevant in our paper, one should make the transformation $|\tilde{n}_i\rangle = \tilde{U}^{(Q)}|n_i\rangle$ instead. In this basis the Hamiltonian can be written as

$$\tilde{H}^{\text{Potts}} = -\sum_{i=1}^{N}\left(\sum_{k=1}^{Q-2}\frac{Q}{Q-1}\tilde{T}_i^k\tilde{T}_{i+1}^{Q-k}\right) + R_i + \lambda P_i + \frac{Q}{Q-1}D_iD_{i+1} + QP_iP_{i+1} - \mathbb{1}_i, \tag{A.3}$$

where the projectors are defined as $P_i = |0\rangle\langle 0|_i$ and $D_i = \mathbb{1}_i - P_i$. The spin-flip operator is modified to $\tilde{T}_i^k|\tilde{n}_i\rangle = (1-\delta_{\tilde{n}_i,0})|1+(\tilde{n}_i-1+k)\bmod(Q-1)\rangle$ and $R_i$ in this basis is given as

$$R_i = \begin{pmatrix} 0 & \sqrt{Q-1} & & & & \\ \sqrt{Q-1} & Q-2 & & & & \\ & & -1 & & & \\ & & & -1 & & \\ & & & & -1 & \\ & & & & & \ddots \end{pmatrix}. \tag{A.4}$$

We are using TeNPy's implementation of the iDMRG algorithms as well as its methods for optimizing tensor contractions exploiting abelian symmetries [59].

Finally we are calculating the ground state of the Potts model for multiple values of the perturbation strength $\lambda$ going as close to criticality as numerically feasible. To decrease the compute time we calculate each groundstate for a specific $\lambda$ by using the previously obtained result of the next higher perturbation strength as initial state to the iDMRG, starting at the product state at $\lambda \to \infty$.

For each value of $\lambda$ the expectation value of the projector $\langle |0\rangle\langle 0|_i\rangle_\lambda$ is evaluated. At $\lambda = 0$ the $Z_Q$ symmetry of the unperturbed Hamiltonian implies an equal expectation value for all $Q$ components, $\langle |0\rangle\langle 0|_i\rangle_{\lambda\to 0} = 1/Q$, thus the order parameter for the Potts model is defined as $\mathcal{O}_m := \langle |0\rangle\langle 0|_i - 1/Q\rangle_\lambda$. By numerically computing the logarithmic derivative we get the exponent

$$\frac{1}{\delta} = \frac{\partial \log \mathcal{O}_m}{\partial \log \lambda}. \tag{A.5}$$

This is done for all values of $Q$ which are of interest and multiple bond dimensions $\chi$ as long as it is numerically feasible, our most sophisticated calculations use $\chi = 2048$ and require up to 3000 sweeps. The results shown in Fig. 1 are converged in $\chi$ and have a negligible error in the numerical derivative.

The iMPS description of a state also allows us to easily obtain the correlation length by analyzing the eigenvalue spectrum of the transfer matrix. The correlation lengths shown in Fig. 1 were extrapolated to infinite bond dimension by the method outlined in Ref. [60].

## B  QMC simulations and exponent estimators

We begin by restating the Hamiltonian in the absence of external fields:

$$H_{JQ} = J\sum_{\langle ij\rangle}(\vec{S}_i \cdot \vec{S}_j - \tfrac{1}{4}) - Q\sum_{\langle ijkl\rangle}(\vec{S}_i \cdot \vec{S}_j - \tfrac{1}{4})(\vec{S}_k \cdot \vec{S}_l - \tfrac{1}{4}). \tag{B.1}$$

The two separate perturbed models are defined by

$$H_{JQ}^m = H_{JQ} + h \sum_i (-1)^{r_i^x + r_i^y + 1} S_i^z \,, \tag{B.2}$$

and

$$H_{JQ}^{\text{vbs}} = H_{JQ} + d \sum_{\langle i,j \rangle \in \hat{x}-\text{even}} (\vec{S}_i \cdot \vec{S}_j - \tfrac{1}{4}) \,. \tag{B.3}$$

Here, in order to keep our formulae as simple as possible, we have opted for the notation $h \equiv \lambda_m$ and $d \equiv \lambda_{\text{vbs}}$.

Beginning with $H_{JQ}^m$, the $h-$field introduces no sign problem since the perturbation is diagonal. However, the field does prohibit the mapping to a deterministic loop model. We therefore simulate the model using the stochastic series expansion (SSE) algorithm [41] with directed loops [42]. On a technical level, we find it convenient to absorb the staggered field into the Heisenberg bond operator, while leaving the $Q-$term in tact. As a result, loop updates on the $Q$ interactions remain deterministic and non-deterministic decisions only need to be made when updating the bond operators.

Directly measuring the Néel order parameter (staggered magnetization) is straightforward in the presence of the external staggered field, since it aquires a nonzero value. More remarkably, the presence of the field allows us to devise a QMC estimator for the effective critical exponent. To define this, we first need to be more explicit about the form of the Hamiltonian, referring the reader to [41] for more general information about the SSE framwork.

First note that the external field only effects diagonal matrix elements of the bond operator, which are given by $\text{diag}(0, \frac{J}{2} + h_B, \frac{J}{2} - h_B, 0)$ in the basis $\{\uparrow\uparrow, \uparrow\downarrow, \downarrow\uparrow, \downarrow\downarrow\}$, and $h_B$ is equal to $h$ divided by the coordination number of the lattice. We have also pulled out an overall minus sign. We now shift the bond operators by $h_B + \epsilon$, so the diagonal part becomes $\text{diag}(h_B + \epsilon, \frac{J}{2} + 2h_B + \epsilon, \frac{J}{2} + \epsilon, h_B + \epsilon)$. Here $\epsilon$ has been introduced to lower the bounce probabilities obtained from solving the directed loop equations, and we have used $\epsilon = 4h$ in our simulations.

Now that we know the matrix elements, we can see that the weights of the QMC configurations are proportional to

$$W(c) \propto \left(\frac{Q}{4}\right)^{N_Q} \left(\frac{J}{2}\right)^{N_J} \left(\frac{J}{2} + \epsilon\right)^{N_{D_0}} \left(h_B + \epsilon\right)^{N_{D_1}} \left(\frac{J}{2} + 2h_B + \epsilon\right)^{N_{D_2}} \,, \tag{B.4}$$

where $N_Q$ is the number of $Q$ matrix elements, $N_J$ is the number of off-diagonal $J$ matrix elements and $N_{D_i}$ are the numbers of different diagonal $J$ matrix elements. Differentiating this weight with respect to $h_B$ gives

$$\frac{\partial W(c)}{\partial h_B} = \left(\frac{N_{D_1}}{h_B + \epsilon} + \frac{2N_{D_2}}{\frac{J}{2} + 2h_B + \epsilon}\right) W(c) \,. \tag{B.5}$$

We can now compute $\partial \mathcal{O}_m / \partial h_B$, which is the main ingredient for the exponent estimator:

$$\frac{\partial \mathcal{O}_m}{\partial h_B} = \frac{\partial}{\partial h_B} \left(\frac{\sum_c m(c) W(c)}{\sum_c W(c)}\right) = \frac{\langle m N_{D_1}\rangle_{\text{C}}}{h_B + \epsilon} + \frac{2\langle m N_{D_2}\rangle_{\text{C}}}{\frac{J}{2} + 2h_B + \epsilon} \,, \tag{B.6}$$

where $\langle m N_{D_i}\rangle_{\text{C}} \equiv \langle m N_{D_i}\rangle - \langle m\rangle\langle N_{D_i}\rangle$ is the "connected" average and $m$ is the staggered $S^z$ magnetization per site. Finally we can write the exponent estimator all together as:

$$\frac{\partial \log(\mathcal{O}_m)}{\partial \log(h)} = \frac{h_B}{\langle m\rangle} \left(\frac{\langle m N_{D_1}\rangle_{\text{C}}}{h_B + \epsilon} + \frac{2\langle m N_{D_2}\rangle_{\text{C}}}{\frac{J}{2} + 2h_B + \epsilon}\right) \,. \tag{B.7}$$

To make our measurements as precise as possible, we can average over the entire imaginary time history when computing the staggered magnetization. This is facilitated by only considering matrix elements that change the staggered magnetization as one moves through the operator sequence.

We also note that simulations and measurements are nearly identical in the Heisenberg bilayer model, where in that case again the field is incorporated into the nearest-neighbor $J_1$ term, and updates on $J_2$ matrix elements are deterministic.

We now describe the measurement of the VBS effective exponent in the $H_{JQ}^{\text{vbs}}$ model, which is slightly more complicated but conceptually similar. The Hamiltonian is given by

$$H_{JQ}^{\text{vbs}} = H_{JQ} + d \sum_{\langle i,j \rangle \in \hat{x}-\text{even}} (\vec{S}_i \cdot \vec{S}_j - \tfrac{1}{4}). \tag{B.8}$$

So the even columns of $x$-bonds have matrix elements $\frac{J}{2} + \frac{d}{2}$, whereas the odd columns of $x$-bonds and all $y$-bonds have matrix elements $\frac{J}{2}$ as normal. This clearly favors one of the four columnar VBS patterns. The associated order parameter is $\mathcal{O}_{\text{vbs}} = \langle \mathcal{P}_{1,2} \rangle - \langle \mathcal{P}_{2,3} \rangle$, where $\mathcal{P}_{i,j} = (\frac{1}{4} - \vec{S}_i \cdot \vec{S}_j)$ is the singlet projector on sites $i, j$. $\mathcal{O}_{\text{vbs}}$ is then just the difference in the expectation value of an even $x$-bond and an odd $x$-bond.

First note that these expectation values can be measured within the SSE framework as

$$\langle \mathcal{P}_{1,2} \rangle = \frac{\langle N_{J_{xe}} \rangle}{\frac{1}{2} N_{\text{site}} \beta (J + d)}, \qquad \langle \mathcal{P}_{2,3} \rangle = \frac{\langle N_{J_{xo}} \rangle}{\frac{1}{2} N_{\text{site}} \beta J}, \tag{B.9}$$

where $N_{J_{xe}}$ ($N_{J_{xo}}$) are the number of even (odd) $x$-bonds in the operator string, which is why we have divided by $N_{\text{site}}/2$ to get the value on a single bond. It is also necessary to divide by $(J + d)$ and $J$, since the number operators give averages of the operators appearing in the Hamiltonian, which are multiplied by those factors. We refer the reader to [61] for useful derivations and formulas for bond operator measurements in the SSE.

As before, we now want to compute the derivative with respect to $d$. We will show how this is done starting with $\langle \mathcal{P}_{1,2} \rangle$:

$$\frac{\partial \langle \mathcal{P}_{1,2} \rangle}{\partial d} = \frac{1}{\frac{1}{2} N_{\text{site}} \beta} \left\{ \frac{\frac{\partial}{\partial d} \langle N_{J_{xe}} \rangle}{J + d} - \frac{\langle N_{J_{xe}} \rangle}{(J + d)^2} \right\}. \tag{B.10}$$

The derivative of $\langle \mathcal{P}_{2,3} \rangle$ is given by:

$$\frac{\partial \langle \mathcal{P}_{2,3} \rangle}{\partial d} = \frac{\frac{\partial}{\partial d} \langle N_{J_{xo}} \rangle}{N_{\text{site}} \beta J}. \tag{B.11}$$

Now in order to compute the derivatives $\frac{\partial}{\partial d} \langle N_{J_{xe}} \rangle$ and $\frac{\partial}{\partial d} \langle N_{J_{xo}} \rangle$, we express the QMC weights as previously. This time the configuration weights are proportional to

$$W(c) \propto \left( \frac{Q}{4} \right)^{N_Q} \left( \frac{J}{2} \right)^{N_{J_y} + N_{J_{xo}}} \left( \frac{J}{2} + \frac{d}{2} \right)^{N_{J_{xe}}}, \tag{B.12}$$

where $N_Q$ is the number of $Q$-operators, $N_{J_y}$ is the number of $y$-oriented $J$ operators, and $N_{J_{xe}}$ ($N_{J_{xo}}$) is the number of $x$-oriented $J$ operators at even (odd) locations from before. The derivative with respect to $d$ is

$$\frac{\partial W(c)}{\partial d} = \left( \frac{N_{J_{xe}}}{J + d} \right) W(c). \tag{B.13}$$

We then have

$$\frac{\partial \langle N_{J_{x\alpha}} \rangle}{\partial d} = \frac{\partial}{\partial d} \left( \frac{\sum_c N_{J_{x\alpha}}(c) W(c)}{\sum_c W(c)} \right) = \left\langle N_{J_{x\alpha}} \left( \frac{N_{J_{xe}}}{J + d} \right) \right\rangle_C, \tag{B.14}$$



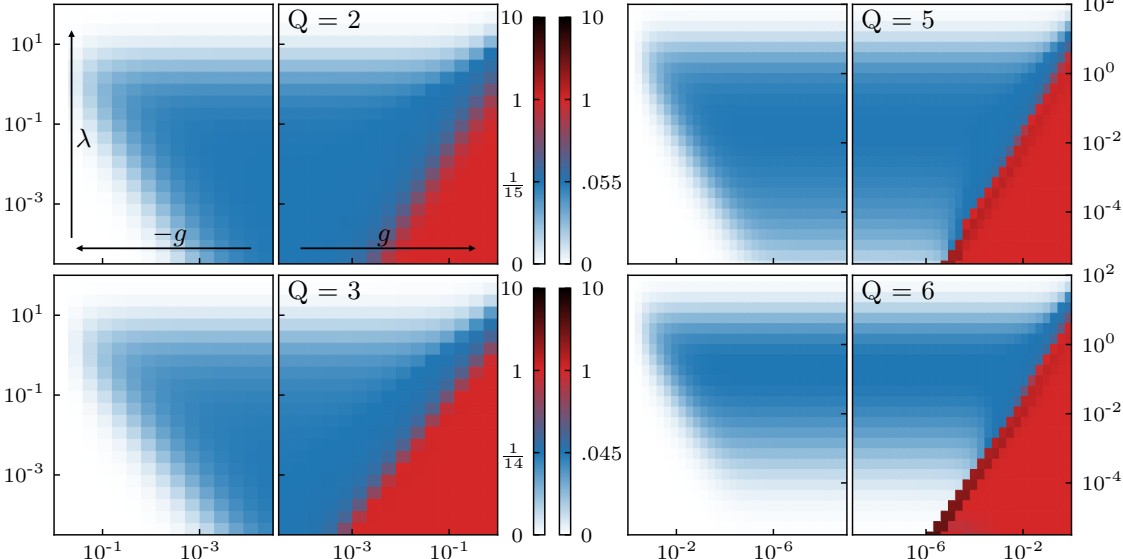

Figure 4: Value of the running exponent $1/\delta$ (colorbar) as function of $\lambda$ and $g$. Shown are two models with continuous transition ($Q = 2, 3$ at $\chi = 64$) and two models with a first order transition ($Q = 5, 6$ at $\chi = 256$). To improve readability the colorbar is adjusted for each model by relating it to values at the phase transition ($g = 0$), where we used the exactly known running exponent and the peak indicated in Fig. 1 for $Q = 2, 3$ and $Q = 5, 6$ respectively.

where $\alpha = e, o$ and again the subscript C means the "connected" part. We can now express

$$\frac{\partial}{\partial d}\Big(\langle \mathcal{P}_{1,2}\rangle - \langle \mathcal{P}_{2,3}\rangle\Big) = \frac{1}{\frac{1}{2}N_{\text{site}}\beta}\left\{ \frac{\langle N_{J_{xe}} N_{J_{xe}}\rangle_{\text{C}} - \langle N_{J_{xe}}\rangle}{(J+d)^2} - \frac{\langle N_{J_{xo}} N_{J_{xe}}\rangle_{\text{C}}}{J(J+d)} \right\}. \tag{B.15}$$

Finally the full expression for the exponent estimator is given by

$$\frac{\partial \log(\mathcal{O}_{\text{vbs}})}{\partial \log(d)} = \frac{d}{\left\langle \frac{N_{J_{xe}}}{J+d} - \frac{N_{J_{xo}}}{J} \right\rangle}\left\{ \frac{\langle N_{J_{xe}} N_{J_{xe}}\rangle_{\text{C}} - \langle N_{J_{xe}}\rangle}{(J+d)^2} - \frac{\langle N_{J_{xo}} N_{J_{xe}}\rangle_{\text{C}}}{J(J+d)} \right\}. \tag{B.16}$$

In the end, it is just necessary to make measurements of $\langle N_{J_{xe}}\rangle$, $\langle N_{J_{xo}}\rangle$, $\langle N_{J_{xe}} N_{J_{xe}}\rangle$, and $\langle N_{J_{xe}} N_{J_{xo}}\rangle$. One can then compute the effective exponent using Eq. (B.16) and the statistical error can be computed by bootstrapping the binned data.

## C Detuning from phase transition

While the running exponent directly at the phase transition (i.e. at $g_c$) of the Potts model has been discussed in detail in Sec. 3 it is interesting to investigate what happens if the system is detuned away from the phase transition, for example because $g_c$ is not known precisely enough. By perturbing the Potts model with a transverse field with coupling strength $g$ as introduced in Eq. (A.1) the system is taken away from criticality, which serves to illustrate the range over which we observe critical scaling as $\lambda \to 0$, here shown in Fig. 4. We find that, as expected, in the continuous cases when $Q = 2, 3$ and when $g < 0$ (favoring the ordered phase), our running exponents drift toward zero as $\lambda \to 0$. One may worry that if this were the case studying a generic model, one might erroneously conclude a first order transition. However

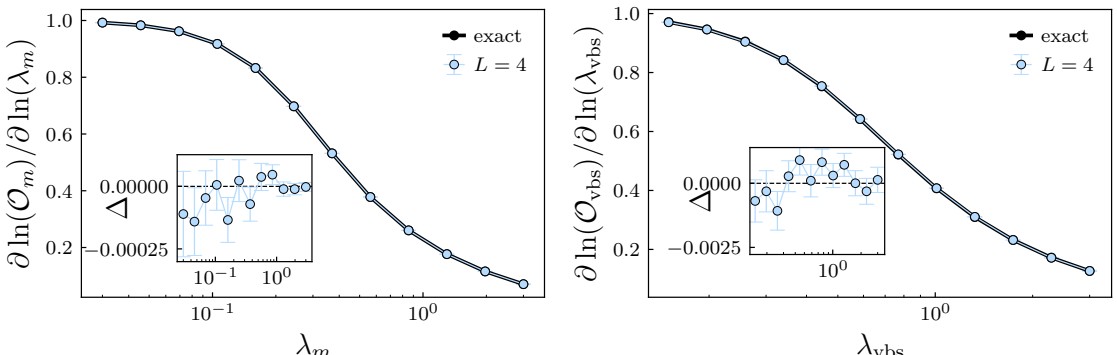

Figure 5: A comparison of the QMC exponent estimators in the $J$-$Q$ model compared with exact results on an $L = 4$ system. We have set $Q = 1$ and $J = 0.0451$ in both cases. The insets show the difference between the QMC data and the exact values.

we see that further increasing $g$ there is a wide swath - akin to a critical fan - where the exponent saturates to a consistent finite value before eventually reaching the linear regime of the disordered phase. This is to be contrasted with the case of $Q = 5, 6$, where the corresponding fan is absent as $\lambda \to 0$. We conclude that when applying our methodology to generic models, it may be necessary to study several transition point estimates to conclude first order behavior. We note that in the case of the $J$-$Q$ model, this subtlety does not appear since we separately study both order parameters at the same transition point.

## D   QMC versus exact diagonalization

In order to confirm the validity of our QMC simulations and exponent estimators, we compare with exact results obtained on small system sizes. Here we focus on the $J$-$Q$ model on an $L = 4$ square lattice. Fig. 5 shows both of the exponent estimators compared to exact diagonalization, where finite differences have been used to compute the logarithmic derivatives. For both the staggered magnetization exponent (left panel) and the VBS exponent (right panel) we have used $J = 0.0451$ and $Q = 1$. In both cases we observe agreement within the QMC error bars, which can be seen in the insets where the difference between the QMC and ED values are plotted.

## E   $J - Q$ model with rectangular lattice anisotropy

A simple way of producing a first-order Néel to VBS phase transition is to introduce rectangular lattice anisotropy into the $J$-$Q$ model, as was previously studied for general $SU(N)$ spin symmetry in [16]. Adopting this same setup, we take spatially anisotropic couplings $J_y/J_x = 0.8$ and $Q_y/Q_x = 0.8$ on rectangular lattices with $L_x = 4L_y/3$. The rectangular lattice anisotropy induces a two-fold degenerate pattern in the VBS phase. We then can estimate the value of the transition based on the binder cumulant of the staggered magnetization, as measured in the pure model without yet introducing the external order parameter fields. The binder cumulant is defined as

$$\mathcal{R}_m = \frac{5}{2}\left(1 - \frac{1}{3}\frac{\langle m_z^4 \rangle}{\langle m_z^2 \rangle^2}\right). \tag{E.1}$$

In Fig. 6 we measure the binder cumulant for different system sizes as function of $J_x/Q_x$, taking $\beta = L_x/2$ in units where $Q_x = 1$. The step in the binder cumulate is an estimate

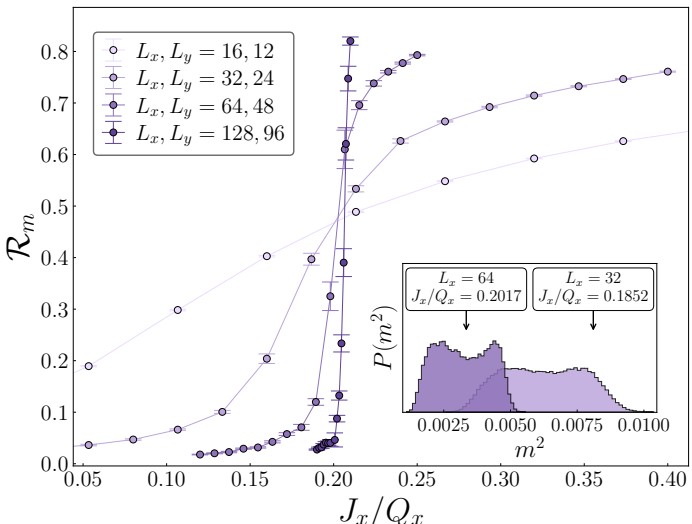

Figure 6: Locating the Néel to two-fold VBS transition in the *J*-*Q* model with rectangular lattice anisotropy. Here we focus on the magnetic signal of the transition by measuring the binder cumulant of the staggered magnetization. A rough estimate of the transition at $J_x/Q_x \approx 0.205$ is more than enough precision for the system sizes used in the main text. We further demonstrate the conventional signs of a first-order transition in this model by showing histograms of our staggered magnetization measurements. A clear double peaked structure emerges with increasing system size near the transition, indicating distinct free energy minima.

for the transition point, which we roughly estimate to be $J_x/Q_x \approx 0.205$. This is more than enough accuracy than is needed for the system sizes used in the main text ($L_x \leq 64$). We note that the first-order nature of the transition is strong enough for us to detect conventional symptoms such as double peaked histograms of our binned staggered magnetization measurements, which are shown in the inset. Comparing histograms near the transition shows the peaks becoming more pronounced with increasing system size, indicating a thermodynamic free energy with distinct local minima.

## F  Zero temperature convergence

We would briefly like to demonstrate the absence of finite temperature effects in the size-independent portion of our QMC data for the running exponents. In all cases we have chosen $\beta \sim L$, with a prefactor larger than the inverse velocity of spin excitations. This ensures that the imaginary time direction grows sufficiently large as a function of $L$ such that only the ground state contributes in the thermodynamic limit. Once the data from different system sizes begins to overlap, we can then be confident that this portion of the the curve is also converged to zero temperature. We demonstrate this for the square lattice *J*-*Q* model in Fig. 7, where we have taken $J = 0.447$ ($Q = 1$) and two values of the inverse temperature ($\beta = L/2$ and $\beta = L$). Here we see that the two data sets only differ in the finite-size regions of the curve, whereas the size independent regions are unaffected.

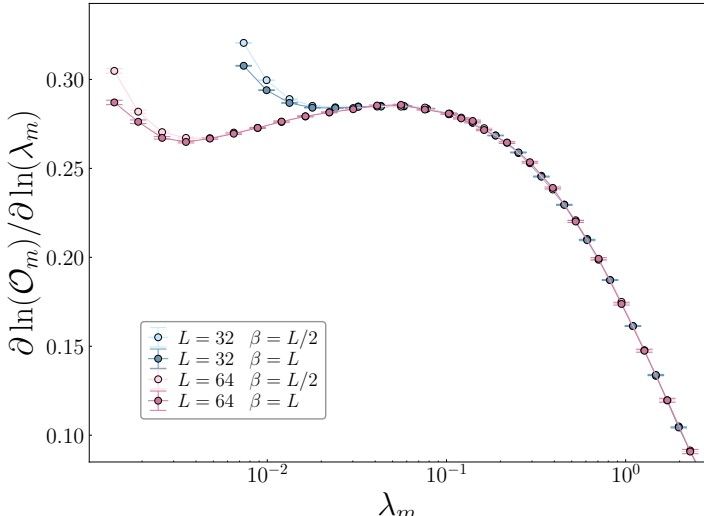

Figure 7: Absence of finite temperature effects in the size-independent region of the running exponents. Here we show data from the square lattice *J-Q* model as a function of the Néel field taken with $J = 0.0447$ ($Q = 1$) at two different inverse temperatures. We note that the data only significantly differs in the finite-size region of the curves (too low field values for a fixed system size), whereas the size independent portion is unaffected.

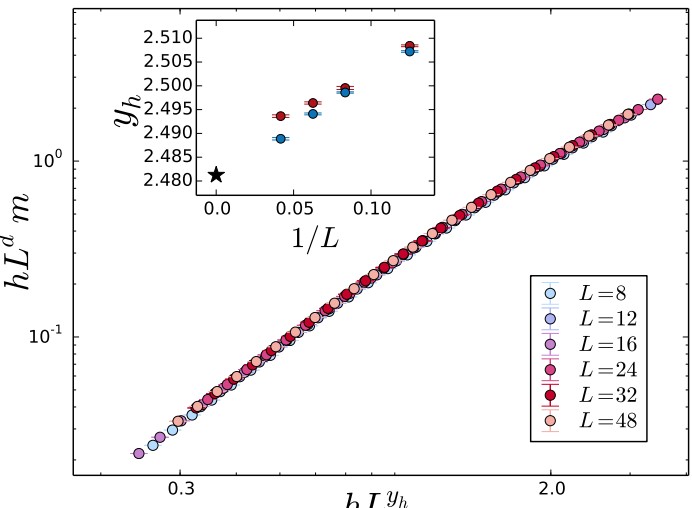

Figure 8: Here we demonstrate critical finite-size scaling of the staggered magnetization as a function of system size and external staggered field with $J_2/J_1 = 2.5223$. Exactly at the critical point, the scaling ansatz is $m = L^{y_h - D} f(hL^{y_h})$, which makes the quantity $hL^D m$ only a function of $hL^{y_h}$. Here $y_h = D/(1/\delta + 1)$. The inset shows the best estimate for $y_h$ based on pair collapses of system sizes $(L, 2L)$ plotted as a function of $1/L$. This procedure was done for both $J_2/J_1 = 2.52205$ and $2.5223$, where the latter shows better agreement with the exponent estimate from the literature [39], and is shown in the main panel collapse. We note that these field values are significantly lower than the ones used in the main text.

# G  Additional bilayer data

Here we present supplementary data for the Heisenberg bilayer near the transition. As can be seen in the main paper, plotting the raw effective exponent as a function of the external field does not provide a high precision determination of the order parameter exponent. In an effort to observe fine-grained resolution of the exponent and to further demonstrate unambiguous critical scaling in this model we perform data collapses [62, 63] at the transition and as a function of the external field and system size, as can be seen in Fig. 8. Here by plotting the exponent obtained from pair collapses of system sizes $(L,2L)$, we observe high sensitivity with respect to the value of the transition used during data collection (shown for $J_2/J_1 = 2.52205$ and 2.5223 in the inset). Of the two values tested, the best agreement with the exponent quoted in the literature [39] is obtained with $J_2/J_1 = 2.5223$ and so this value is used in the data presented in the main text.

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
