# Peer review of "Diagnosing weakly first-order phase transitions by coupling to order parameters"

_SciPost Physics, doi:SciPost Phys. 15, 061 (2023)_

## Round 1 · Referee Report · Johannes Hauschild (Referee 1) · 2022-3-15

Strengths

  • clear presentation of the proposed method
  • application of the method to a debated transition with clear indications to a first-order transition, illustrating the potential for further applications.

Weaknesses

  • assumption that already fine-tuned to the critical point

Report

Numerical simulations trying to resolve a weakly first-order quantum phase transition are plagued by the large correlation lengths at the criticial point, rendering a finite-size scaling analysis very challenging. The authors suggest to circumvent this difficulty by explicitly coupling the system to the (symmetry-breaking) order parameter $O$, i.e. $H -> H + \lambda O$. While $\langle O\rangle \propto \lambda ^{1/d}$ as $\lambda \rightarrow 0$ at a continuous phase transition, for a first order transition $ \langle O \rangle \rightarrow m > 0 $. The authors demonstrate that this behaviour can be clearly distinguished by studying the "running exponent" $\frac{d \log \langle O \rangle}{d \log \lambda}$ as a function of $\lambda$. Crucially, these signatures appear already at intermediate values of $\lambda$, where the correlation length can be much smaller than directly at the transition, rendering numerical simulations better controlled. The authors test this method on the quantum Potts models using (infinite) DMRG and the O(3) transition in variants of the Heisenberg model with quantum monte carlo, specifically the stochastic series expansion (SSE). As a technical detail, they develop estimators for the running exponent in SSE. Finally, the authors revisit the transition from Neel order to a valence bond solid in the J-Q-model, the nature of which is still debated in literature. Coupling to any of the two order parameters, their SSE results using this method are in clear favor of a first order transition.

By revisiting the original definition of the order parameter under an applied field and employing it for numerical simulations, the manuscript gives a breakthrough for distinguishing weakly forst-order transitions from continuous ones, with a lot of potential follow-up works. Further, all general acceptance criteria are met.

I only have two questions/comments to the authors:

1) The method of the authors assumes that the Hamiltonian is already fine-tuned to the critical point (as clearly stated in the paper). However, in many practical cases, the exact location of the transition is not necessary known a priori. Hence, I was wondering what happens if the method is applied slightly away from the critical point, to either side of the transition. My expectation would be: in the symmetry broken phase, $\langle O \rangle$ would approach a finite value as $\lambda \rightarrow 0$, such that a continuous transition might show similar signatures as a first-order transition - do you agree in that respect? If my intuition is right, I think it might be beneficial for future application to caution the reader and discuss this in the paper or at least an appendix.

2) Do you see a way to apply the method to transitions from/to topological phases? It seems difficult if there is no local order parameter, or am I missing something here?

Requested changes

1) Discuss what happens if the method is applied slightly away from the critical point, see report. 2) Add "University of Waterloo" and/or URI to Ref. 59.

---

## Round 1 · Referee Report · Anonymous (Referee 2) · 2022-3-23

Strengths

(i) The paper is very well-presented, reads very smoothly, and is clear.
(ii) The numerical approach presented, to the best of my knowledges, seems to be accurate, and is tested against known behavior.
(iii) The work seems to answer an outstanding question in the community, namely that nature of the phase transition from a Neél antiferromagnet to a valence bond solid is first-order.

Weaknesses

I would not call it a weakness, but I am not sure how this method can be employed in cases when the critical point is not known exactly. And I could not find this point addressed by the authors, though I think it would be beneficial to do so.

Report

The paper deals with a numerical approach utilizing what the authors call a running exponent. The latter is a log-derivative of the order parameter at the critical point with respect to a symmetry-breaking field strength $\lambda$. In the limit of $\lambda\to0$, depending on whether or not the transition if continuous or first-order, the running exponent will go to zero in the latter case, while acquiring a unique finite value in the former case. This stems from the fact that at first-order transition points, the order parameter is not zero, while for continuous phase transitions, it is zero at the critical point.

They successfully test their method confirming established behavior in the Potts model using infinite matrix product state techniques. The major achievement of this work is that it seems to settle a hitherto unanswered question regarding the nature of the phase transition between Néel order and VBS. Applying their approach to the J-Q model using quantum Monte Carlo methods, they reveal that the relevant order parameters are finite at this transition, which is strong evidence for a first-order phase transition.

Requested changes

It would be beneficial if the authors addressed how they would deal with models where the transition point is not precisely known.

---

## Round 1 · Referee Report · Anonymous (Referee 3) · 2022-4-8

Report

D'Emidio, Eberharter and Lauchli analyse quantum phase transitions, suggesting to distinguish weakly first order transitions and continuous transitions by studying the effect of smaller and smaller external order-parameter fields. It is well-known that, at a continuous transition point, the induced order parameter expectation value is proportional to a power of the external field while it converges to a constant value at first order transitions.

The behaviour of the corresponding running exponent is determined for the Q-state Potts model using iMPS, for the O(3) transition in three two-dimensional models with different microscopic interactions, and for the Neel to valance-bond-solid transition in the J-Q model. The external field explicitly breaks symmetries which reduces the efficiency in numerical simulations and requires taking double limits with respect to system size and the external field instead of just extrapolating in the system size. On the other hand, the authors argue that the double limit is no substantial obstacle and that the running exponent provides a clearer signal than, say, cumulants or susceptibilities in symmetry-preserving simulations.

The paper is written very clearly and provides insightful numerical data. I have a few questions that could be addressed in an updated version: - It is argued that using external fields is more efficient than the symmetry-preserving method. It might be helpful to show corresponding (inconclusive) data from the symmetric approach if possible. For the Q-state Potts model and the O(3) transition, probably both methods would work equally well. The case of the J-Q models seems more intricate. While the data in Fig. 3 is rather suggestive in favour of a first order transition it might still be possible for the running exponent to converge to a nonzero value. - The simulations used previously determined values for transition points. If these were not very precise and actually lying in the symmetry broken phase of a continuous transition, wouldn't the running exponent look as if the transition was first-order? - It is argued that double limits with respect to system size and the external field are no major concern. Is this still true for Monte Carlo and tensor network state simulations with limited accuracy? Typically tensor network state simulations for the relevant two-dimensional models are limited to rather small bond dimensions and biased by choosing a certain elementary cell of tensors. It seems that the resulting errors would pose a challenge for the proper double limit.

Overall, I think that this is a beautiful paper that makes a useful contribution for the study of quantum phase transitions, and I recommend publication in SciPost.

It is not so obvious whether it is appropriate for SciPost Physics which asks for "groundbreaking" and "breakthrough" results. Given that - the use of external fields is basically a textbook approach for the study of phase transitions that has been applied before, - that the distinction between continuous and weakly first-order is of theoretical but relatively limited experimental or technological interest, - and that the addressed models are rather well-known I'd see this more on the side of a regular SciPost article.

---

## Round 2 · Referee Report · Johannes Hauschild · 2023-3-23

Report

I'm satisfied by the authors responses to the questions and the additional Appendix C and recommend a publication.

---

## Round 2 · Referee Report · Anonymous · 2023-3-23

Report

In the response, the authors have addressed all points raised in the referee reports and added data illustrating how the nature of a phase transition can be determined with the suggested approach even if the transition point is previously unknown. While it is at this point not entirely clear how efficient and how widely applicable the approach is compared to others, it is certainly an interesting option and the paper makes a strong point. I recommend publication in SciPost Physics.

---

## Round 2 · Author Response

We thank the referees for their overall positive assessment of our paper and for recommending it for publication in SciPost Physics after minor revision. We are also grateful for the detailed referees comments, which we will clarify below, point by point.

Response to Report 1:

The method of the authors assumes that the Hamiltonian is already fine-tuned to the critical point (as clearly stated in the paper). However, in many practical cases, the exact location of the transition is not necessary known a priori. Hence, I was wondering what happens if the method is applied slightly away from the critical point, to either side of the transition. My expectation would be: in the symmetry broken phase, $\langle O \rangle$ would approach a finite value as $\lambda \to 0$, such that a continuous transition might show similar signatures as a first-order transition - do you agree in that respect? If my intuition is right, I think it might be beneficial for future application to caution the reader and discuss this in the paper or at least an appendix.

Indeed our method requires a prior value for the critical point, although we found that locating it (e.g. by using binder ratio crossings as we did for the rectangular J-Q model) to the required precision was rather easy in this case. It is true that one may worry about being detuned from the transition and falsely claiming a first-order transition, when really it is just the ordered phase. To address this we have included such a study for the Q-state Potts model in the Appendix C with a new figure 4, which helps to show the critical and first-order behavior in a larger parameter space near the transition. One notices that the running exponents shows a qualitatively different behaviour in the parameter plane: a fanlike structure in the continuous case, and a horizontal rod or bar-like structure in the first order case. We believe that more generally it is a good idea to consider several values of the transition as a function of the field before diagnosing a first order transition, although we note that such a subtlety is not present in the J-Q case, since we have ordered phases on both sides.

Do you see a way to apply the method to transitions from/to topological phases? It seems difficult if there is no local order parameter, or am I missing something here?

We have not explored an application to topological phase transition, where there is no local order parameter on either side of the transition. It is likely that our ideas are not applicable for such transitions.

Discuss what happens if the method is applied slightly away from the critical point, see report. 
 Done as detailed above. Add "University of Waterloo" and/or URI to Ref. 59. Thanks for pointing this out, we have updated the corresponding reference.

Response to Report 2:

I would not call it a weakness, but I am not sure how this method can be employed in cases when the critical point is not known exactly. And I could not find this point addressed by the authors, though I think it would be beneficial to do so.

Indeed our method requires a prior value for the critical point, although we found that locating it (e.g. by using binder ratio crossings as we did for the rectangular J-Q model) to the required precision was rather easy in this case. It is true that one may worry about being detuned from the transition and falsely claiming a first-order transition, when really it is just the ordered phase. To address this we have included such a study for the Q-state Potts model in the Appendix C with a new figure 4, which helps to show the critical and first-order behavior in a larger parameter space near the transition. One notices that the running exponents shows a qualitatively different behaviour in the parameter plane: a fanlike structure in the continuous case, and a horizontal rod or bar-like structure in the first order case. We believe that more generally it is a good idea to consider several values of the transition as a function of the field before diagnosing a first order transition, although we note that such a subtlety is not present in the J-Q case, since we have ordered phases on both sides.

Response to Report 3:

It is argued that using external fields is more efficient than the symmetry-preserving method. It might be helpful to show corresponding (inconclusive) data from the symmetric approach if possible. For the Q-state Potts model and the O(3) transition, probably both methods would work equally well. The case of the J-Q models seems more intricate. While the data in Fig. 3 is rather suggestive in favor of a first order transition it might still be possible for the running exponent to converge to a nonzero value.

We remark that while the literature on symmetry preserving studies of these models is extensive, it is true that we do not provide a side-by-side comparison of both the symmetric and perturbed approaches. However, we find such a comparison to be beyond the scope of that which we hope to accomplish here. In the symmetric J-Q setup, it is already well known that the critical exponents show a drift with system size, which have been interpreted as unconventional corrections to scaling. It is our view that the perturbed setup produces a visibly more striking effect that is more easily interpretable as a first order transition, at smaller system sizes than the symmetric simulations.

We agree that it is impossible to exclude nonzero convergence of the running exponents, but this argument is always present in any extrapolation scheme, and it is based on the appearance of a distinct scaling form emerging at inaccessible system sizes.
We do not detect any deviation away from a steady drift of the exponents toward zero on the (rather large) system sizes that we have studied. Given the similarity with the Potts model, where the first order nature is known analytically, we believe a saturation or upturn at much smaller perturbation scales would constitute new, unexpected physics. While we can not rule it out, there is also no mechanism known to us which could give rise to such a saturation or upturn. So the simplest explanation of the observed behaviour remains a first order transition.

The simulations used previously determined values for transition points. If these were not very precise and actually lying in the symmetry broken phase of a continuous transition, wouldn't the running exponent look as if the transition was first-order?

Indeed our method requires a prior value for the critical point, although we found that locating it (e.g. by using binder ratio crossings as we did for the rectangular J-Q model) to the required precision was rather easy in this case. It is true that one may worry about being detuned from the transition and falsely claiming a first-order transition, when really it is just the ordered phase. To address this we have included such a study for the Q-state Potts model in the Appendix C with a new figure 4, which helps to show the critical and first-order behavior in a larger parameter space near the transition. One notices that the running exponents shows a qualitatively different behaviour in the parameter plane: a fanlike structure in the continuous case, and a horizontal rod or bar-like structure in the first order case. We believe that more generally it is a good idea to consider several values of the transition as a function of the field before diagnosing a first order transition, although we note that such a subtlety is not present in the J-Q case, since we have ordered phases on both sides.

It is argued that double limits with respect to system size and the external field are no major concern. Is this still true for Monte Carlo and tensor network state simulations with limited accuracy? Typically tensor network state simulations for the relevant two-dimensional models are limited to rather small bond dimensions and biased by choosing a certain elementary cell of tensors. It seems that the resulting errors would pose a challenge for the proper double limit.

It is hard for us to predict the behaviour or our method when used with Monte Carlo results of poor accuracy. Most likely results are not conclusive and one should improve the accuracy of the Monte Carlo results. Regarding the tensor network simulations, there is a recent proof of principle application of the method proposed in the current manscript in section III.C of the paper: J. Hasik et al., Phys. Rev. B 106, 125154 (2022). In there the authors studied a different microscopic implementation of the O(3) quantum phase transition and compared the iPEPS results to finite size QMC results. Indeed one has to carefully check that the results are converged in bond dimension D for a given value of the perturbation, but for intermediate to large values of the perturbation the scheme worked well in that application. Comparing these converged values of the perturbation with Fig. 3 of our manuscript, an iPEPS study of the JQ model might be within reach.

the use of external fields is basically a textbook approach for the study of phase transitions that has been applied before

We agree (and also state in the manuscript) that the approach as such is text book material. We stress however that it has been overlooked that this approach has an unappreciated strength in detecting weakly first order transition, and we use our finding to rule the transition in the J-Q model to be first order.

the distinction between continuous and weakly first-order is of theoretical but relatively limited experimental or technological interest

Of course the technological prospects of our work is quite limited indeed, but Neel-VBS transitions on the Shastry Sutherland lattice are currently being discussed in the context of experiments on Strontium Copper Borate (SCBO).

the addressed models are rather well-known

We find this adds to the importance of our work, since the nature of this well studied transition was still disputed in the community before our work.

---

## Round 2 · List of Changes

Minor adjustments were made to formatting and wording throughout the document.

We added emphasis on locating transition point of the rectangular J-Q model in section 5 on page 10.

We added appendix C “Detuning from phase transition” to address a point made by alle 3 referees. This section includes a new figure, Fig. 4, which illustrates the difference in behaviour for continuous and first order phase transitions upon detuning from the phase transition in the case of the Q-state Potts model.

---

## Editorial Decision

published